# Over-coupled resonator for broadband surface enhanced infrared absorption (SEIRA)

Laura Paggi[1], Alice Fabas[1], Hasnaa El Ouazzani[1], Jean-Paul Hugonin[2], Nikos Fayard[2], Nathalie Bardou[3], Christophe Dupuis[3], Jean-Jacques Greffet [2] & Patrick Bouchon [1] ✉

Detection of molecules is a key issue for many applications. Surface enhanced infrared absorption (SEIRA) uses arrays of resonant nanoantennas with good quality factors which can be used to locally enhance the illumination of molecules. The technique has proved to be an effective tool to detect small amount of material. However, nanoresonators can detect molecules on a narrow bandwidth so that a set of resonators is necessary to identify a molecule fingerprint. Here, we introduce an alternative paradigm and use low quality factor resonators with large radiative losses (over-coupled resonators). The bandwidth enables to detect all absorption lines between 5 and 10 μm, reproducing the molecular absorption spectrum. Counterintuitively, despite a lower quality factor, the system sensitivity is improved and we report a reflectivity variation as large as one percent per nanometer of molecular layer of PMMA. This paves the way to specific identification of molecules. We illustrate the potential of the technique with the detection of the explosive precursor 2,4-dinitrotoluene (DNT). There is a fair agreement with electromagnetic simulations and we also introduce an analytic model of the SEIRA signal obtained in the over-coupling regime.

The development of detection technologies is a key issue for many societal challenges such as health, sustainable development, security, and defense. Infrared spectroscopy is a well-known cost-effective and non-destructive technique enabling the identification of chemical species thanks to their vibrational resonances[1]. Their spectrum form a fingerprint. Unfortunately, small molecules interact only slightly with long-wavelength infrared light and show a weak signal in this range. Surface-Enhanced Infrared Absorption (SEIRA) spectroscopy takes advantage of the fact that absorption is proportional to the squared electric field to increase molecular absorption in the vicinity of metallic structures supporting collective electronic oscillations, i.e., plasmonic modes[2]. It has been widely investigated since it was first demonstrated in 1980 with gold and silver thin films[3]. The development of optical

nanoresonators with localized hot spots has enabled SEIRA sensors with improved performances[4–15]. They have been applied to the detection of species for biosensing[7,14–23], environmental monitoring with gas sensing[24–26] or to other domains linked to industrial control[27], or defense applications[13]. All the previous works are based on the same paradigm: IR absorption can be enhanced by first enhancing the incident field using a resonator and then inserting the analyte assuming that its presence will not perturb the resonator. The highest field enhancement demonstrated has been obtained thanks to the use of nanometer-sized gaps down to 3 nm, leaving very limited room for further improvement[28–31].

A downside to these cited structures is that they all operate on a narrow wavelength range, restraining the coupling to only one or a

[1]DOTA, ONERA, Université Paris-Saclay, Palaiseau, France. [2]Laboratoire Charles Fabry, Institut d'Optique Graduate School, CNRS, Université Paris-Saclay, Palaiseau, France. [3]Centre de Nanosciences et de Nanotechnologies (C2N), CNRS, Université Paris-Saclay, Palaiseau, France. ✉ e-mail: patrick.bouchon@onera.fr

very limited number of vibration modes. Besides, the insertion of an analyte in the vicinity of the resonator also induces a surface plasmon resonance (SPR) effect that further complexifies the analysis of the SEIRA signal. The multiplexed approach has been suggested with resonators spanning the spectral band of interest, but it relies on matricial sensors[14,32]. Designs to enhance a larger portion of the spectrum with multi-resonant structures that could work with a single-pixel detector have also been proposed[23,27,33–37], but most of them have achieved weak SEIRA signals compared to previous demonstrations or signals resulting from coupled SEIRA and SPR effects. The integration of a broadband epsilon near zero material has also been suggested theoretically, but epsilon near zero effects have been so far experimentally demonstrated on very narrow spectral ranges[38]. Designing broadband sensitive SEIRA detectors is still an unresolved issue.

Through the description of resonator-analyte coupling with temporal coupled mode theory (TCMT), Adato et al. highlighted the importance of considering the coupled system as a whole to optimize the SEIRA signal and not only take a high field enhancement as the unique criterion in the design of resonators[39]. The critical coupling configuration of a structure, giving the maximum absorption is obtained when the ratio between radiative and non-radiative losses is equal to unity. Most of the time, resonators are designed in critical coupling or under-coupling and tuned with one or a few absorption lines. However, the insertion of an absorbing material in the resonator unbalances this ratio and the SEIRA enhancement is no longer optimal.

In this letter, we introduce an over-coupled resonator that is able to achieve beyond state-of-the-art SEIRA signal on a very broad spectral range while being free of the SPR effect. For the sake of comparison, we first use PMMA, which is a material that has been widely studied for SEIRA demonstrations. We demonstrate experimentally the ability to enhance the absorption of many lines over a broad spectrum from 5 to 10 μm, and report reflectivity differences up to 0.7 for 45 nm thick film and 0.25 for 9 nm thick film which also reproduce the molecular absorption spectrum. The SEIRA signal in the over-coupling regime is analyzed in the framework of the temporal coupled mode theory, and we demonstrate that a given coupling ratio leads to the highest SEIRA signal. We then show that the reflectivity difference can be linked with a linearized model to the amount of absorbing material that is in agreement with experimental measurements. We finally apply the technique to the detection of 2,4-dinitrotoluene (DNT).

## Results and discussion
### SEIRA in an over-coupled resonator
The resonator is composed of a dielectric cavity surmounted by a metallic layer, structured with a periodic array of slits where the electric field is highly confined at resonance. This type of structure has been previously studied by Fabas et al.[13]. Here, the dielectric layer is taken to be zinc sulfide (ZnS), which has the advantage of being transparent in the mid-infrared. The structure can be described as an LC oscillator, with the slit acting as a capacitor and the dielectric layer as an inductance similar to the resonator studied in Chevalier et al.[40–42] (Supplementary Fig. 1). We can tune the resonance by changing the geometrical parameters $h_{Au}$, $h_{ZnS}$, $w$, and $d$ in order to optimize the coupling with an analyte and obtain the maximum enhancement. A schematic of the structure is represented in Fig. 1a.

Figure 1b shows the calculated reflectivity of a resonator with parameters $h_{Au} = 45$ nm, $h_{ZnS} = 50$ nm, $w = 50$ nm, and $d = 765$ nm with and without PMMA. The slits are filled with PMMA and a continuous layer of PMMA of thickness $h_{Au}$ is placed on top of the gold ribbons. The bare resonator is at critical coupling (i.e., $R = 0$) at $\lambda = 5.41$ μm. It is tuned to match the strong PMMA vibration mode at 1730 cm$^{-1}$ (5.78 μm) of the carbonyl group after the addition of PMMA. As shown in the inset, the electric field intensity is enhanced in the volume of the slit at resonance by a factor of 300. Calculations of field enhancement for

various configurations are shown in Supplementary Fig. 2. It is clearly seen in Fig. 1b that the reflectivity variation is due to two different effects. On the one hand, the presence of an analyte introduces a shift of the reflectivity dip so that the reflectivity at a given wavelength changes, this is referred hereafter as surface plasmon resonance (SPR) mechanism. It mostly depends on the real part of the analyte permittivity. On the other hand, the antenna enhances the absorption of the analyte absorption lines so that it contains information on the absorption fingerprint of the polymer deposited on the resonator, this is the classical SEIRA effect with a shift due to SPR[13]. The coupling of a bright plasmonic mode to a dark molecular vibration mode gives rise to a dip of 23%. A similar behavior was described in Adato et al.[39].

By increasing the thickness $h_{ZnS}$, the device can switch to the over-coupled regime. The reflectivity of such a resonator is shown in Fig. 1c. The period has been slightly shifted (d = 600 nm) so that the wavelength of resonance becomes $\lambda = 6.67$ μm. The system is no longer at critical coupling so that the reflectivity dip does not reach 0. The enhancement of the electric field intensity in the slit at resonance is multiplied by approximately 1.7 compared to the critically coupled case (maps are available in Supplementary Fig. 2).

One could expect that when filling the slit with PMMA, a similar SPR effect is obtained with a slightly better SEIRA signal due to the enhancement of the electric field intensity. Surprisingly, the spectrum in the presence of PMMA is drastically different from both the spectra of the bare resonator and of the critically coupled resonator in the presence of PMMA. Firstly, all the vibrational modes of PMMA between 5 and 10 μm are strongly enhanced, and secondly, for each of them, very high contrast values are obtained. While both resonators have the same slit volume where the electric field enhancement occurs, the over-coupled resonator gives rise to higher values of SEIRA signal on a broader range.

### SEIRA analytic model using the coupled-mode theory
This behavior can be understood when considering the coupled mode model of absorption for a resonator as a function of the frequency $\omega$[39]:

$$A = \frac{4\gamma_r \gamma_{nr}}{(\omega - \omega_r)^2 + (\gamma_r + \gamma_{nr})^2} \tag{1}$$

where $\gamma_r$ and $\gamma_{nr}$ are the radiative and non-radiative losses, and $\omega_r$ is the resonance frequency. A typical SEIRA resonator at critical coupling with large enhancement has $\gamma_r = \gamma_{nr}$ to ensure critical coupling and a large quality factor $\gamma_r \ll \omega_r$. By increasing $\gamma_r$ ($\gamma_r \gg \gamma_{nr}$) to get in the over-coupled regime, the critical coupling condition is no longer satisfied and the quality factor is reduced so that the reflectivity dip is smaller but broader. This enables to probe lines over a large bandwidth since the cavity enhancement effect operates for $(\omega - \omega_r) \sim \gamma_r$. We show in the Supplementary Equations (3–5) that the addition of absorbing material results in a modified absorptivity:

$$A = \frac{4\gamma_r(\gamma_{nr} + \gamma_\mu)}{(\omega - \omega_r - \omega_\mu)^2 + (\gamma_r + \gamma_{nr} + \gamma_\mu)^2} \tag{2}$$

where $\omega_\mu = \frac{\mu^2(\omega - \omega_b)}{(\omega - \omega_b)^2 + \gamma_b^2}$ and $\gamma_\mu = \frac{\mu^2 \gamma_b}{(\omega - \omega_b)^2 + \gamma_b^2}$ are the shift and decay induced by the absorber, expressed in terms of $\omega_b$ the central frequency of the absorption line, $\gamma_b$ its linewidth, and $\mu$ the interaction parameter between the resonator and the absorption line. Equation (2) shows that the absorbing material brings back the system towards the critical coupling ($\gamma_r = \gamma_{nr} + \gamma_\mu$) in a narrow spectral range defined by the resonance condition $\omega = \omega_r + \omega_\mu$. This results in a large reflectivity dip for each absorbing line in the over-coupled resonator bandwidth, that we quantify considering the difference of reflectivity with and without the absorbing material: $\Delta R(\omega) = |r_{without}(\omega)|^2 - |r_{with}(\omega)|^2$. In Supplementary Fig. 7, we study analytically the function $\Delta R(\omega)$ in the limit of an

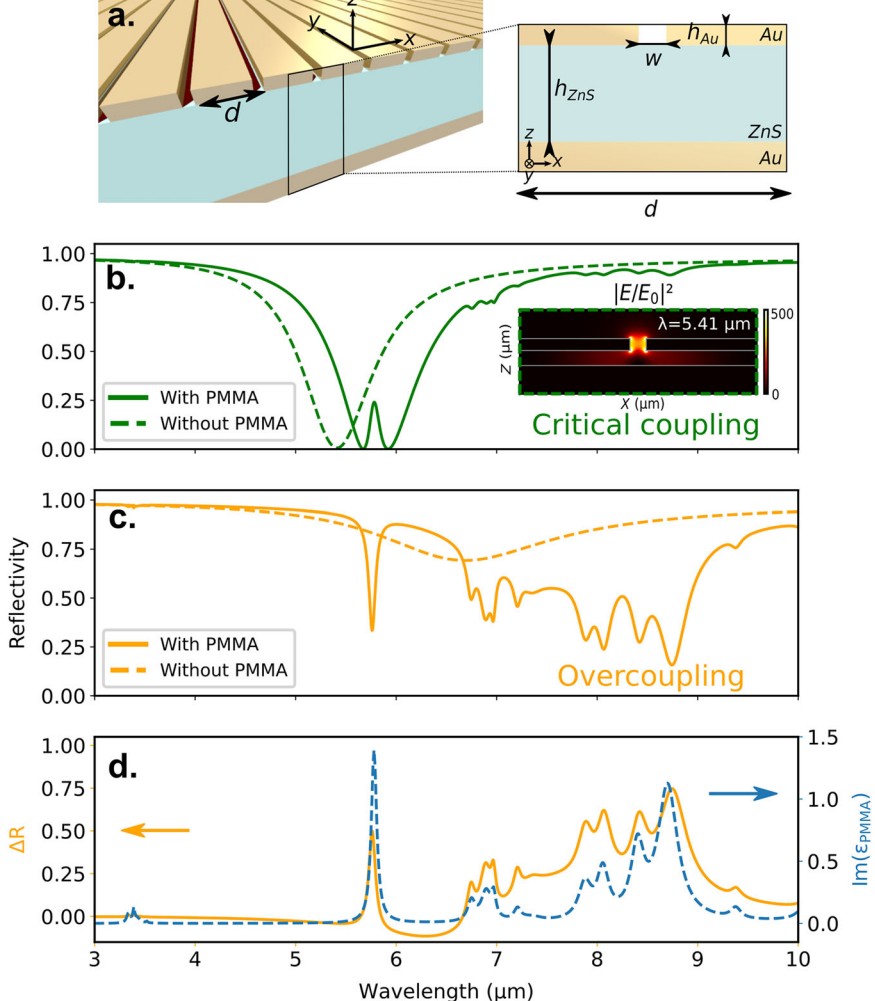

**Fig. 1 | Metasurface for SEIRA of PMMA in the critical coupling and over-coupling configuration. a** Scheme of the resonator array (period $d$). Calculated reflectivity spectra of **b** a critically coupled resonator (green lines) with dimensions $h_{Au} = 45$ nm, $h_{ZnS} = 50$ nm, $w = 50$ nm and $d = 765$ nm, and **c** an over-coupled resonator (orange lines) with dimensions $h_{Au} = 45$ nm, $h_{ZnS} = 280$ nm, $w = 50$ nm and $d = 600$ nm with (continuous line) and without (dashed line) a 45 nm thick PMMA layer filling the slits. **d** Calculated reflectivity difference $\Delta R$ spectrum of the over-coupled resonator (orange continuous line) and imaginary part of the dielectric permittivity of PMMA (blue dashed line).

absorption line far detuned for the resonator: $(\omega_b - \omega_r) \gg \gamma_b$. We show that $\Delta R(\omega)$ is maximal for a slightly shifted value of the molecule resonance frequency $\omega = \omega_b + \frac{\mu^2(\omega_b - \omega_r)}{(\omega_b - \omega_r)^2 + (\gamma_r + \gamma_{nr})^2}$ and derive the expression of its maximum value that we denote $\Delta R_m$. We plot in Fig. 2b the variation of $\Delta R_m$ as a function of $f = \gamma_r/\gamma_{nr}$ and observe the existence of a maximum of the SEIRA signal for $f_{max} = (\omega_b - \omega_r)/\gamma_{nr} \gg 1$ deep inside the over-coupled regime. This derivation demonstrates that over-coupled resonators are more sensitive to absorption lines far detuned from the resonator than under-coupled and critically coupled resonators.

Usually, SEIRA effect is quantified post-treatment, using, for instance, a smoothing algorithm or artificial materials to decouple SEIRA from SPR effect[13,31]. Remarkably, in the over-coupled configuration, the SPR effect has less weight on the reflectivity difference $\Delta R = R_{withoutPMMA} - R_{withPMMA}$ that can be directly used to identify and quantify the chemical species. The reflectivity difference is plotted in Fig. 1d for the over-coupled resonator and compared to the imaginary part of the PMMA dielectric permittivity. We see that all vibrations are clearly identifiable without the need for an additional or intermediary treatment. The slight shift of the absorption lines is explained by the model, $\omega$ is faintly blueshifted when $(\omega_b - \omega_r) < 0$ and redshifted when $(\omega_b - \omega_r) > 0$. In summary, over-coupled configuration demonstrates

stronger enhancement than critical coupling, over a large wavelength range of $5\,\mu m$, with more direct and clearer results that simplify the data analysis.

## Experimental demonstration of SEIRA in over-coupled resonators

To observe experimentally these ideas, we fabricated resonators with sections of the slits of $50 \times 45$ nm$^2$. The 280 nm thick ZnS layer has been deposited on top of a gold substrate with a 5 nm chromium adhesion layer. The 45 nm gold ribbons were created by electron beam lithography (EBL) and a lift-off using PMMA resist with a 5 nm chromium adhesion layer. A scanning electron microscope image is shown in Fig. 3. The infrared response is measured by Fourier Transform Infrared Spectroscopy with a Vertex 70 from Bruker coupled to a Hyperion 2000 microscope. The Cassegrain objective used has a numerical aperture of 0.4 and all the measurements are normalized by the reflectivity of a gold mirror. Reflectivity spectra are represented in Fig. 3b. Under illumination by transverse magnetic (TM) polarized infrared beam, with the magnetic field parallel to the slits, the large and weak resonance appears at $5.7\,\mu m$. There is no resonance in transverse electric (TE) polarization. The small peak at $5.8\,\mu m$ indicates that a PMMA residue is still present in the slits after the lift-off.

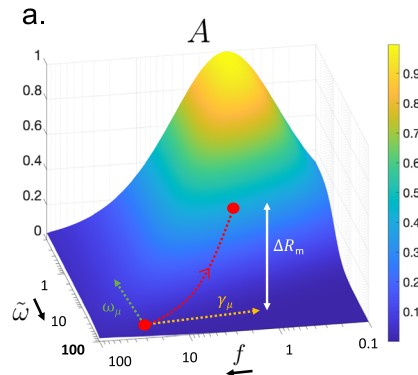

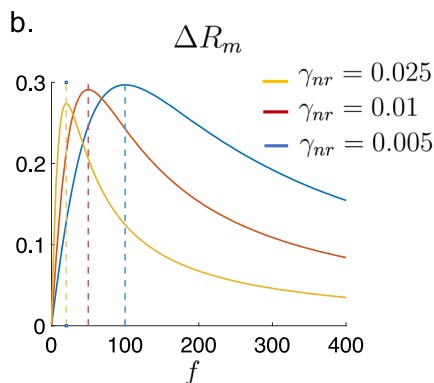

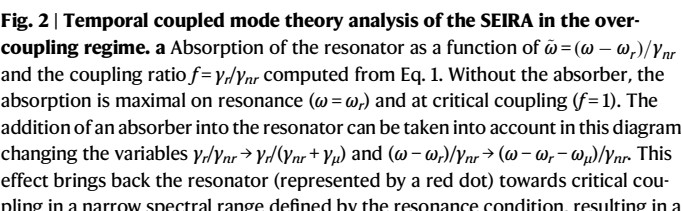

**Fig. 2 | Temporal coupled mode theory analysis of the SEIRA in the over-coupling regime. a** Absorption of the resonator as a function of $\tilde{\omega} = (\omega - \omega_r)/\gamma_{nr}$ and the coupling ratio $f = \gamma_r/\gamma_{nr}$ computed from Eq. 1. Without the absorber, the absorption is maximal on resonance ($\omega = \omega_r$) and at critical coupling ($f = 1$). The addition of an absorber into the resonator can be taken into account in this diagram changing the variables $\gamma_r/\gamma_{nr} \rightarrow \gamma_r/(\gamma_{nr} + \gamma_\mu)$ and $(\omega - \omega_r)/\gamma_{nr} \rightarrow (\omega - \omega_r - \omega_\mu)/\gamma_{nr}$. This effect brings back the resonator (represented by a red dot) towards critical coupling in a narrow spectral range defined by the resonance condition, resulting in a

potentially large reflectivity difference $\Delta R_m$. **b** Reflectivity difference $\Delta R_m$ as a function of $f = \gamma_r/\gamma_{nr}$ computed in Supplementary Fig. 8. To vary $f$, we fix $\gamma_{nr}$ and let $\gamma_r$ vary. $\Delta R_m$ is plotted for $\gamma_{nr} = 0.025$ (yellow line), $\gamma_{nr} = 0.01$ (red line), and $\gamma_{nr} = 0.005$ (blue line). In the dashed vertical line, we plot the value $f_{max} = (\omega_b - \omega_r)/\gamma_{nr}$ associated to each value of $\gamma_{nr}$ to highlight the position of the maximum of $\Delta R_m(f)$. The normalized parameters are : $\omega_r = 1$, $\omega_b = 1.5$, $\gamma_r = 0.5$, $\gamma_b = 0.01$ and $\mu = 0.03$.

We demonstrate the enhancement potential of the structure by filling entirely the slits with PMMA. To do so, we spin coat A2 PMMA solution on the resonator in order to obtain a 45 nm thick layer. Under TM polarization, we observe a very broad enhancement of the different absorption peaks of the molecule. As expected, the TE response is equivalent to a flat mirror with a thin layer of molecules whose response is very weakly enhanced and undetectable with standard IR spectroscopy.

We now discuss the sensitivity with the thickness of the layer. For a given geometry including now a thicker ZnS layer (530 nm), pushing the device further into the over-coupling regime, Fig. 4 shows the evolution of four vibrational modes of PMMA between 5 and 10 μm as a function of the thickness of a continuous ideal PMMA layer $h_{PMMA}$ in the slit. The enhancement of the 1730 cm$^{-1}$ (5.78 μm) mode first increases linearly and then tends to saturate, while the others follow a linear growth with the absorber layer thickness. This linear behavior can be used to directly retrieve the amount of molecules. We show in the Supplementary Equations (19–21) that the resonator behavior can be described by a linearized model as a function of the parameter $(\epsilon - 1)\frac{h_{PMMA}}{h_{Au}}$. The comparison of the model with numerical results is shown in Supplementary Fig. 9. For small values, the reflectivity difference is linearly dependent on this parameter.

Thanks to the observed linear behavior, we define detection responsivity in terms of reflectivity variation per nanometer. In this case, we found that it ranges from 1%/nm to 3%/nm. A reflectivity contrast up to 27% is obtained experimentally with the thinnest PMMA layer estimated to 8.8 nm, which compares to Adato et al. with the exception that the signal is only attributed to the SEIRA effect and not intertwined with the SPR shift. The 1149 cm$^{-1}$ (8.7 μm) mode leads at the same time to a 12% reflectivity contrast.

This detection responsivity could also be improved by reducing the width of the slits, but it would imply a trade-off between the interaction volume and the enhancement bandwidth as well as changes in the fabrication process. An example of design is given in Supplementary Fig. 3.

Finally, we illustrate the ability of the over-coupled regime to identify an absorption fingerprint by applying it to the detection of 2,4-dinitrotoluene (DNT) molecule, a precursor of the explosive

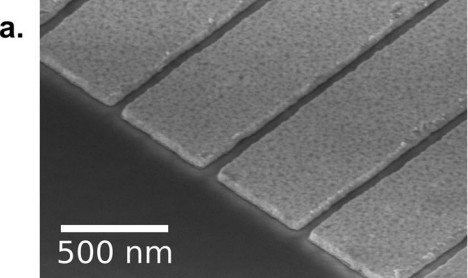

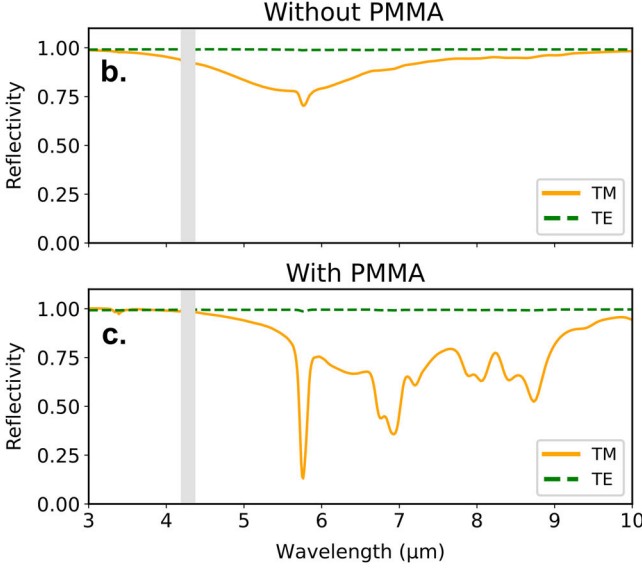

**Fig. 3 | Experimental demonstration of the SEIRA effect. a** Electron beam microscopy image of the edge of the resonator array with dimensions $h_{Au} = 45$ nm, $h_{ZnS} = 280$ nm, $w = 50$ nm, and $d = 600$ nm. Measured spectra of **b** the bare array and **c** the array covered with a 45 nm PMMA layer, for transverse magnetic polarization (orange continuous lines) and transverse electric polarization (green dashed lines). Gray areas cover the absorption of $CO_2$.

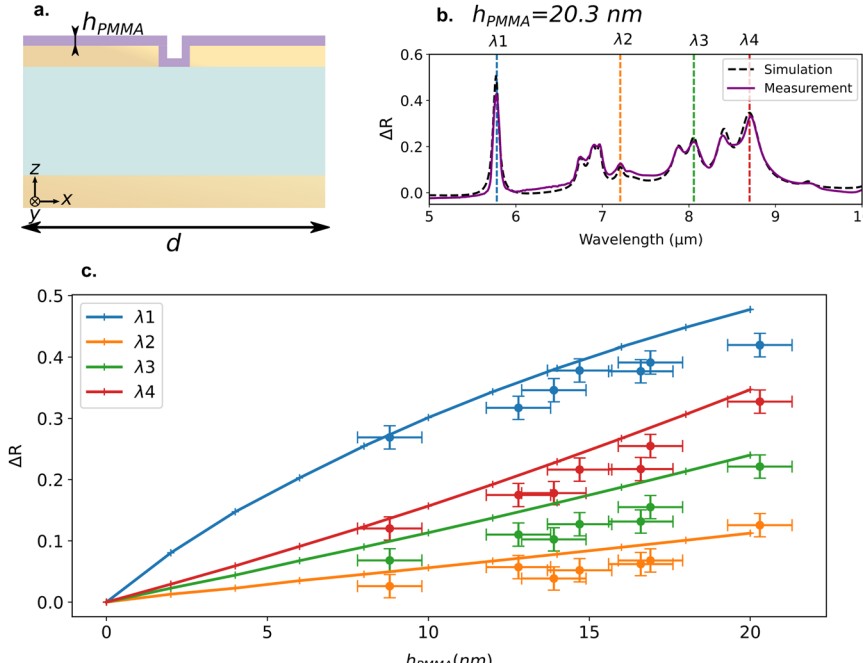

**Fig. 4 | Linear regime of the SEIRA signal in the over-coupled regime for PMMA.**
**a** Schematic of a period of a resonator array covered with an ideal continuous PMMA layer. **b** Measured (dashed black line) and simulated (continuous purple line) reflectivity difference of a resonator (dimensions: $h_{Au} = 45$ nm, $h_{ZnS} = 530$ nm, $w = 50$ nm and $d = 300$ nm) covered with a 20 nm continuous PMMA layer in the slit and on top of the ribbons. **c** Simulated reflectivity difference of four vibration modes (indicated as $\lambda 1$, $\lambda 2$, $\lambda 3$, and $\lambda 4$ in **b**) as a function of the PMMA layer thickness up to 20 nm. Experimental dots are also plotted for various PMMA thicknesses measured with an ellipsometer. Error bars correspond to standard deviation.

molecule trinitrotoluene. DNT was diluted in acetonitrile with a concentration of 1.5 g/L. A microfluidic printer Microplotter II by Sonoplot was used to deposit a controlled volume of solution over a desired surface of $500 \times 500\,\mu m^2$ with a micropipette of $20\,\mu m$ diameter tip. The solvent evaporates in a few seconds and the slits are then filled with DNT molecules. The measured resonator array is identical to the structure used in Fig. 4, illustrating the versatility of the resonator. The infrared fingerprint of DNT can be recovered three minutes after deposition (Fig. 5a) with reflectivity differences up to 50% for the 1538 cm$^{-1}$ (6.5 μm) resonance. In this case again, several absorption peaks are clearly recognizable on a large wavelength range (between 5 and 10 μm). The experimental spectrum can be fitted considering a Drude-Lorentz model for DNT[13] and an effective uniform layer of 60 nm on top of the resonators and filling the slits. The DNT molecule is an organic volatile compound, thus the signature of thin layers disappears as it evaporates. To illustrate its volatility, Fig. 5b presents the measured spectrum four minutes after deposition. The reflectivity difference has decreased to 20% for the stronger vibration modes. To retrieve numerically the measurements, a DNT layer of 20 nm in the slits is used. The detection responsivity in the case of DNT is close to 1%/nm for the two strongest absorption line. Additional details on the deposition method and measurements are given in Supplementary Table 2 and Supplementary Figs. 4, 5. Once again the reflectivity difference is linearly related to the imaginary part of the DNT Drude-Lorentz permittivity model.

We have demonstrated that the same over-coupled resonator geometry is able to enhance the absorption of DNT and PMMA between 5 and 10 μm. The broad and weak resonance of the over-coupled structure enables to exploit directly reflectivity difference spectra. Moreover, for thicknesses below 20 nm, we can retrieve quantitative information thanks to a linearized model of the absorption with regards to the quantity of molecules probed. For a majority of the excited vibration modes, the enhancement is also amongst the largest values reported in the state of the art. These slits arrays ensure

with a simple fabrication process, large interaction volumes, very good angular tolerance[13], and easily tunable resonances that could pave the entire fingerprint region with a limited number of resonators. These are clear advantages for the design and integration of compact and versatile infrared spectroscopy sensors.

## Methods
### Simulation and design
Calculations were obtained using a B-Spline Modal Method and confirmed with RCWA modal method[43,44]. The ZnS layer permittivity is modeled thanks to a modified Sellmeier model taken from Klein et al.[45]. The gold layers are modeled using a Drude model: $\epsilon(\lambda) = 1 - \frac{1}{(\frac{\lambda_p}{\lambda} + i\gamma)\frac{\lambda_p}{\lambda}}$, with $\lambda_p = 159$ nm and $\gamma = 0.0077$ to fit the literature data in the infrared[46]. Chromium and titanium adhesion layers are modeled with Brendel-Bormann models from Rakic et al.[47]. The PMMA permittivity is given by a Drude-Lorentz model adapted from Tsuda et al.[48]: $\epsilon(\omega) = \epsilon_\infty + \sum_n A_n \frac{\omega_{0,n}^2}{\omega_{0,n}^2 - \omega^2 + i\gamma_{L,n}\omega}$, where $\epsilon_\infty = 2.162$ is the dielectric constant at high frequency, $A_n$ is the amplitude of the $n^{th}$ oscillator, $\omega_{0,n}$ is its resonance frequency and $\gamma_{L,n}$ its damping factor. The exact parameters values are available in Supplementary Table 1. The 2,4-dinitrotoluene layers are also modeled with a Drude-Lorentz function with parameters taken from Fabas et al.[13]. To faithfully simulate measurements obtained with the Cassegrain objective (Numerical aperture 0.4), reflectivity spectra are calculated and averaged over incident angles between 12° and 24° and azimuthal angles between 0° and 90° and divided by the spectrum of a gold mirror used as a reference.

### Fabrication
The resonators introduced in the first experimental results in Fig. 3 are obtained by depositing a 200 nm gold layer with a 20 nm Ti adhesion layer on a silicon wafer, and then a 280 nm ZnS layer is deposited. The results shown in Fig. 5 are done with a 530 nm thick ZnS layer.

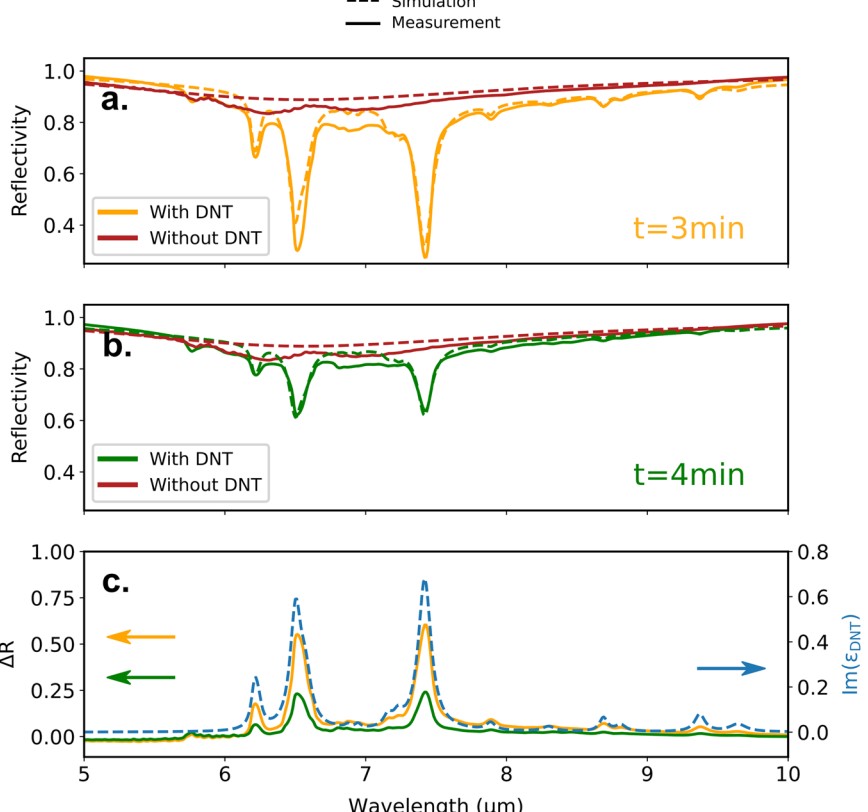

**Fig. 5 | SEIRA effect on 2,4-dinitrotoluene. a** Measured (dashed lines) and calculated (continuous lines) reflectivity spectra of the previously introduced resonator with dimensions $h_{Au} = 45$ nm, $h_{ZnS} = 530$ nm, $w = 56$ nm, and $d = 300$ nm measured at $t = 3$ min (yellow lines) after deposition of DNT molecules and **b** $t = 4$ min (green lines) after deposition. The spectra on the bare resonator without molecules are plotted for comparison (solid red line for simulation and dashed red line for measurement). Continuous lines correspond to experimental data and dashed lines to fitted calculations. **c** Measured reflectivity difference spectra $\Delta R$ 3 min after deposition (solid orange line) and 4 min after deposition (solid green line) with respect to the left axis, and imaginary part of the dielectric permittivity of DNT model (blue dashed line)[13] with respect to the right axis.

To pattern the gold ribbons, PMMA A3 resist was spin-coated at 3000 rounds/min for 60 s and then baked at 170 °C for 5 min. Then, e-beam lithography was done with a Vistec EBPG 5000 at 100 kV with a proximity effect correction that accounts for the electron scattering in the layers stack. The resist is developed in a solution of MIBK:IPA (1:3) for 70 s and rinsed in IPA for 10 s. Evaporation of 40 nm of gold with a 5 nm Cr adhesion layer was performed and the substrate was lifted in SVC-14 at 80 °C. The first array used for PMMA experiments in Fig. 2b. presents a small peak at 5.8 μm due to the presence of residual traces of PMMA from the lift-off in the active volume of the slit. The mode is enhanced by the resonator and therefore visible in the spectrum of the bare array. The resonators fabricated for the following experiments, especially with DNT do not demonstrate a peak as visible as an additional ion etching step was added in the fabrication process before deposition of the gold top layer to remove and reduce traces of resist.

### Measurement procedure

The sample is placed under a Bruker Hyperion microscope which is coupled to an FTIR spectrometer Bruker Vertex 70. The illumination is provided by a focused Globar source and shines on the device with an incident angle between 12° and 24°. The detector is a mono-element MCT with a cut-off wavelength at 16 μm that needs to be cooled with liquid nitrogen. A Cassegrain objective with numerical aperture NA = 0.4 is used to focus the beam. The background is acquired on a gold substrate accounting for the back side gold mirror. The spectra are averaged over 64 scans with a 6 cm⁻¹ spectral resolution in the first experiments with PMMA with a $100 \times 100$ μm² diaphragm. In order to

obtain a uniform layer of 45 nm filling the slits, A2 PMMA resist was deposited by spin-coating at 5000 rounds/min and then baked for 5 min at 170 °C. Other thicknesses are obtained with A2 PMMA diluted in anisole and various depositions speed, as described in Supplementary Table 2. Thicknesses were monitored with ellipsometric measurements performed next to the arrays, the surface was investigated with Atomic Force Microscopy images and homogeneity of the infrared response was evaluated across the $1 \times 1$ mm² arrays (Supplementary Figs. 4, 5). In the case of 2,4-dinitrotoluene, the compound was purchased from Sigma Aldrich and diluted in acetonitrile with a concentration of 1.51 g/L. A microfluidic printer Microplotter II by Sonoplot with a glass micropipette of 20 μm diameter aperture is used to deposit about $(2.0 \pm 0.4) \times 10^{-2}$ μL, that corresponds to $(30.2 \pm 6.8)$ ng of DNT on the resonator on a surface of $500 \times 500$ μm². After solvent evaporation in a few seconds, only the targeted molecules are present in the resonator. Measurements are realized within five minutes before complete evaporation of DNT molecules and spectra are averaged over 32 scans with a 6 cm⁻¹ spectral resolution.

## Data availability

Datasets supporting the findings of this study are available in Zenodo with the identifiers https://doi.org/10.5281/zenodo.7858261(https://zenodo.org/record/7858261).

## Code availability

The code used in this work is available in the same Zenodo repository as the datasets.

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

## Acknowledgements
This work was Funded by the Agence Nationale de la Recherche grant Dartagnan (ANR-20-CE39-0003). This work was done within the C2N micro nanotechnologies platforms and partly supported by the RENA-TECH network and the General Council of Essonne. We acknowledge Jules Lackner for ellipsometer measurements.

## Author contributions
P.B. and J.J.G. conceived and supervised the project. L.P., A.F., and H.E.O. made the experimental characterizations. N.B. and C.D. made fabrication of the samples with inputs from P.B. J.P.H. designed the devices with inputs from A.F. and L.P. and formalized the homographic approximation. N.F. and J.J.G. developed the TCMT for over-coupled configurations. L.P., J.J.G., and P.B. wrote the paper with inputs from all authors.

## Competing interests
The authors declare no competing interests.
