## [Peer Review File · Nature Communications]

Over-coupled resonator for broadband surface enhanced infrared absorption (SEIRA)REVIEWER COMMENTS

Reviewer #1 (Remarks to the Author):

In this work, the authors describe a plasmonic system composed of a dielectric cavity where one mirror is replaced with an array of metallic wires and apply it for surface-enhanced infrared absorption spectroscopy (SEIRA). The notable features of the authors' design are that it is operated in the overcoupled light-matter interaction regime and functions over a relatively broad spectral range. Simple SEIRA demonstrations based on thick layers of PMMA and ODT molecules are likewise presented.

Even though the proposed approach is described clearly and appears (in principle) useful for SEIRA applications, the manuscript fails to demonstrate significant advances over the current state of the art. My main concerns are with (1) the claimed sensitivity advances over previously published geometries, (2) the usefulness of the broadband operation, and (3) the lack of a compelling SEIRA demonstration with a relevant molecular system. Furthermore, the mechanism of the enhancement is not analyzed in sufficient detail and some data necessary for following the authors' arguments is missing. These concerns preclude publication of the manuscript in Nature Communications, as explained in more detail below.

As the authors state themselves, the different coupling regimes of plasmonic SEIRA have been extensively studied in the past 10 years, starting with the Adato et al. papers mentioned in the text. Since then, the field has progressed and many demonstrations of specially engineered plasmonic antennas have been shown for the SEIRA-based detection of ultralow quantities of molecules (see, e.g., ACS Photonics 5, 4117–4124 (2018)). These antennas still exhibit relatively broad resonances, which can resolve many different types of biomolecules simultaneously (e.g., Adv. Mater. 33, 2006054 (2021)). It is not clear how the authors' design improves over these structures in terms of sensitivity. The molecular sensing demonstrations (PMMA, ODT) are done with extremely thick layers of molecules (50-100 nm), which greatly exaggerates the sensor response in the spectra. From the performed calculations and experiments, it is nearly impossible to draw a comparison to realistic molecular sensing experiments (such as on low concentrations of molecules or ultrathin layers). This issue becomes especially clear in the comparison table in Figure S2. The authors show a reflectance modulation of "10-70%" for their own design and a PMMA thickness of 100nm. This is objectively worse than the Adato 2013 demonstration, where a PMMA layer of only 8 nm thickness produced a modulation of 25%.

Likewise, broadband SEIRA operation with high sensitivity has been conclusively demonstrated in a recent paper using wavelength-multiplexed plasmonic hook nanoantennas (Nature Communications 13, 3859 (2022)), where "the wavelength-multiplexed HNAs serve as ultrasensitive vibrational probes in a continuous ultra-broadband region (wavelengths from 6 μm to 9 μm)."

The published paper in fact goes much beyond the present work by demonstrating molecular recognition from different alcohol mixtures using sophisticated machine learning algorithms. As an alternative, many recent SEIRA demonstrations employ a multiplexed approach (for example through arrays of sensor elements), which can also produce broadband operation and high sensitivity.

Finally, the authors omit some information that would be crucial for better understanding the enhancement mechanism of their design. For example, only the critically coupled and overcoupled regimes are shown. Many other approaches operate in the undercoupled regime. How those the authors' design perform there? What is the general mechanism of the broadband operation? Why is the electromagnetic near-field not maximal for the critical coupling case (as expected from theory and literature)? What are the mechanisms and limits of the spectral operating range?

In summary, I believe that the manuscript does not demonstrate a significant advance for plasmonic SEIRA (especially with the lack of a compelling sensing demonstration) and therefore recommend its

rejection.

Reviewer #2 (Remarks to the Author):

Paggi et al. investigates broadband, high-contrast SEIRA sensing of thin films in a previously overlooked 'over-coupled' regime.

In the majority of SEIRA studies reported to date, as the authors correctly state, a common design approach has been to enhance IR absorption in bare resonator/antenna structures and then to insert target analytes in and around optical hotspots.

The authors approach is different in that they utilized an over-coupled resonator structure consisting of an array of gold ribbons coupled to a gold mirror. While the reflectivity spectrum of the bare resonator shows weak IR absorption, interestingly the structure shows broadband SEIRA signals from deposited PMMA and DNT thin films. This counter-intuitive outcome is nicely explained by using analytical equations and computer simulations.

The manuscript is clearly written and suggests a promising new approach for the SEIRA sensing community, so I recommend publication in Nature Communications after the authors address the following comments:

1. Before delving into the discussion on the critical vs. over-coupled regimes, it'd be nice to add more materials to explain the 'mechanics' of their resonator structure and how the geometrical parameters (gap width, ZnS thickness, ribbons periodicity, etc.) influence the overall IR absorption and transition from under- to critical- and over-coupled regimes.

The basic geometry of their resonator structure is reminiscent of a patch antenna structure. Also, similar motifs have been used by other researchers (albeit mostly in critically coupled regime), for example, the gapped gold antennas on a reflector [L. Dong et al. Nano Lett. 2017, 17, 5768-5774] and reflector-coupled gold ribbon arrays separated by a mid-IR transparent cavity [I.-H. Lee et al. Nature Nanotechnology 2019, 14, 313]. Thus, it would be nice to present the authors' design in the context of such relevant previous works.

2. If it makes sense to improve the flow, the author might consider moving the field enhancement simulation (Figure S1) into the main text.

3. Figure 3 PMMA results:

- Labels 'a', 'b', 'c', 'd' are mentioned in the caption but are missing in the figure. Please add proper labels.

- In (a) schematic, the authors depict ideal coverage of 45-nm thick PMMA on the film and inside the gap, which is unrealistic. Has the authors considered non-uniform coverage of the structure after PMMA coating? Could there be more (thicker) PMMA in and around the slit area?

- More experimental spectra could be shown in Figure 3.

4. Likewise, when the authors discuss the SEIRA results obtained from DNT in Fig. 4, I was hoping to find more details on how they prepared and characterized the sample after depositing a droplet. In the Supplementary Information, the authors infer that ~100 nm thick DNT film should have been formed, but more independent measurement would be desirable. Was the coverage very uniform across the active surface area?

5. Could the authors discuss strategies to further optimize the performance? Their simulations show the E-field intensity enhancement ($|E|^2$) of 300x in their structure. By shrinking the gap width, it may be possible to further enhance the field enhancement factor. Could that lead to improved SEIRA sensitivity/contrast in the over-coupled regime?

6. In their 'Conclusion', it'd be nice to go beyond their current structure and discuss if the strategy of performing SEIRA in the over-coupled regime can be generalized. For example, could the authors suggest general design rules for other SEIRA structures (e.g. nanorods) based on what they've learned here?

7. In Figure 4, reflectivity different (y-axis), is the unit supposed to be %?

First, we want to thank both reviewers for their detailed reading of our manuscript and for the questions and remarks that has helped us to significantly improve the quality of our manuscript, which we have now complemented both in the main manuscript and the supplemental material after completing further studies.

Please, find below the details of our response to their comments and the actions which we have taken to modify our manuscript accordingly.

Reviewer 1

General comment: “In this work, the authors describe a plasmonic system composed of a dielectric cavity where one mirror is replaced with an array of metallic wires and apply it for surface-enhanced infrared absorption spectroscopy (SEIRA). The notable features of the authors’ design are that it is operated in the overcoupled light-matter interaction regime and functions over a relatively broad spectral range. Simple SEIRA demonstrations based on thick layers of PMMA and ODT molecules are likewise presented.

Even though the proposed approach is described clearly and appears (in principle) useful for SEIRA applications, the manuscript fails to demonstrate significant advances over the current state of the art. My main concerns are with (1) the claimed sensitivity advances over previously published geometries, (2) the usefulness of the broadband operation, and (3) the lack of a compelling SEIRA demonstration with a relevant molecular system. Furthermore, the mechanism of the enhancement is not analyzed in sufficient detail and some data necessary for following the authors’ arguments is missing. These concerns preclude publication of the manuscript in Nature Communications, as explained in more detail below.”

Response: We reply below to the detailed concerns of the reviewer. In particular, we have introduced in our revised manuscript new results on more relevant molecular systems (thinnest layers: 8 nm for PMMA and 20 nm for DNT) and developed the mechanism of the enhancement thanks to a model based on the temporal coupled mode theory.

Comment 1 :

(1) the claimed sensitivity advances over previously published geometries

As the authors state themselves, the different coupling regimes of plasmonic SEIRA have been extensively studied in the past 10 years, starting with the Adato et al. papers mentioned in the text. Since then, the field has progressed and many demonstrations of specially engineered plasmonic antennas have been shown for the SEIRA-based detection of ultralow quantities of molecules (see, e.g., ACS Photonics 5, 4117–4124 (2018)). These antennas still exhibit relatively broad resonances, which can resolve many different types of biomolecules simultaneously (e.g., Adv. Mater. 33, 2006054 (2021)).

It is not clear how the authors’ design improves over these structures in terms of sensitivity. The molecular sensing demonstrations (PMMA, ODT) are done with extremely thick layers of molecules (50-100 nm), which greatly exaggerates the sensor response in the spectra. From the performed calculations and experiments, it is nearly impossible to draw a comparison to realistic molecular sensing experiments (such as on low concentrations of molecules or ultrathin layers). This issue becomes especially clear in the comparison table in Figure S2. The authors show a reflectance modulation of “10-70%” for their own design and a PMMA thickness of 100nm. This is objectively worse than the Adato 2013 demonstration, where a PMMA layer of only 8 nm thickness produced a modulation of 25%.

Response: We thank the reviewer for the additional references and we have added them to our revised manuscript.

The reviewer has pointed out an important issue regarding the fairness of the comparison between geometries. The comparison between systems that are demonstrated with different molecules is complex as the strength of the molecules absorption is paramount to the response of the complete system resonator + molecules. And this is what motivated our choice for PMMA, since as shown in the comparison table in supplemental material, SEIRA has been demonstrated on many geometries with this molecule with thicknesses ranging from 8 to 100 nm.

As requested, in our revised manuscript, we have made new experiments to demonstrate the sensitivity for thinner layers between 8 to 20 nm for PMMA and for 20 nm for DNT.

Besides, the sensitivity of the resonator can be improved by shrinking the size of the slit, as is demonstrated numerically in the following figure (added as Fig. S3). By reducing the dimensions of the slits to 10 nm by 8 nm (width x height), one can obtain a reflectivity difference up to 50 % for the 1730 cm^{-1} PMMA mode when the slit is filled by the analyte.

In the original manuscript, we show experimentally a reflectance modulation of 10-70% on a PMMA thickness of 45 nm and we have also introduced the responsivity of the resonator as being in the order of 1-3%/nm with numerical simulations. This gives a value of 24 % for an 8 nm thick layer, which seems comparable to the value we have indicated for the Adato 2013 demonstration.

But this value of 25% indicated for the Adato paper is due to both the SEIRA effect and the SPR (surface plasmon resonance) effect. may be confusing as on one hand it should have been underlined that it is obtained with calculated spectra, the experimental reflectivity difference value is lower, on another hand it is not clear that the reflectivity difference is induced by the enhancement of the mode only and not impacted by the shift of the resonance. The following figure shows the superposition of the Adato spectrum and the signature of PMMA, and underlines the complexity recovering the seira signal.

Redacted

It must be emphasized that, for our design, vibrational modes of PMMA at higher wavelength are still visible, the broadband effect is preserved for lower slit volume.

Action taken by the authors:

- We have added in the introduction of the revised manuscript *Adv. Mater.* 33, 2006054 (2021) and *ACS Photonics* 5, 4117–4124 (2018)).

- New results on thinner layers of PMMA and DNT have been added (Figs. 4-5)

- Electromagnetic computations for SEIRA on a smaller slit has been introduced (Fig. S3 and Sec. 1.4 of Supp. Mat.)

Comment 2 : (2) the usefulness of the broadband operation

Likewise, broadband SEIRA operation with high sensitivity has been conclusively demonstrated in a recent paper using wavelength-multiplexed plasmonic hook nanoantennas (Nature Communications 13, 3859 (2022)), where “the wavelength-multiplexed HNAs serve as ultrasensitive vibrational probes in a continuous ultra-broadband region (wavelengths from 6 μm to 9 μm).” The published paper in fact goes much beyond the present work by demonstrating molecular recognition from different alcohol mixtures using sophisticated machine learning algorithms. As an alternative, many recent SEIRA demonstrations employ a multiplexed approach (for example through arrays of sensor elements), which can also produce broadband operation and high sensitivity.

Response: The authors agree with the referee that the mentioned articles must be included as citations to complete the state of the art in broadband seira. The accomplished multiplexing approach, as stated, can lead to 3 μm wide enhancement range. The structure presented in our manuscript has the advantage of enhancing over a 5 μm wide wavelength range with only one resonator. Additionally, our signal reproduces the molecular absorption spectrum and is free of the SPR effect. In both cases, the system can operate with a single pixel detector. The third alternative discussed by the reviewer is to use arrays of sensors with multiplexed approach, and this can be done on an even broader range. This third approach may address other needs, but is not directly comparable to the two first.

We believe that our system can also present advantages in given situations compared to the Nature Comms 13, 3859 (2022) paper. Juxtaposed systems are probing the response of different sets of molecules, while in our case, the same molecules are probed. This could be interesting when

studying dynamic behaviors of molecules, in particular for biological molecules (e.g., following DNA or proteins interactions).

Action taken by the authors:

- We have added the reference Nature Communications 13, 3859 (2022) given by the reviewer on the multiplexed approach, as well as one reference on the multiplex approach with a sensor array in the introduction of our revised manuscript.

Comment 3: (3) the lack of a compelling SEIRA demonstration with a relevant molecular system

Response: As suggested by the reviewer, we have included in our revised manuscript new experimental results on PMMA and DNT. On PMMA, we have made various thicknesses ranging from 8 nm to 20 nm (the thickness of our resonator).

2,4-dinitrotoluene is an explosive derivative, that was chosen to offer a practical demonstration. It is a volatile compound and we have made further measurements for a 20 nm layer.

Action taken by the authors:

- We have added results to the revised manuscript with more relevant systems (see Figs. 4 and 5). Also, the results on DNT are now done with the same resonator materials as with PMMA (ZnS instead of SiO₂).

Comment 4: Furthermore, the mechanism of the enhancement is not analyzed in sufficient detail and some data necessary for following the authors' arguments is missing.

Finally, the authors omit some information that would be crucial for better understanding the enhancement mechanism of their design. For example, only the critically coupled and overcoupled regimes are shown. Many other approaches operate in the undercoupled regime. How those the authors' design perform there? What is the general mechanism of the broadband operation?

What are the mechanisms and limits of the spectral operating range?

Reponse:

We agree with the referee that a more general discussion about the mechanism of the broadband operation would improve the understanding of the experimental results presented in this manuscript. Hence, we used the Temporal Coupled Mode Theory to derive an analytical expression of the reflection coefficient in presence of an absorbing line (equation 2 of the main text) that enables the study of the variation of the amplitude of the SEIRA signal as a function of the different parameters of the system.

In particular we show that for a good metal, *i.e.*, $\gamma_{nr} \ll (\omega_r - \omega_b)$:

- the radiative decay rate γ_r dictates the operating bandwidth of our method. Indeed, the frequency response of the absorbing material has to overlap with the frequency response of the resonator to be detected. From this observation, we directly know that under-coupled, and critically coupled resonators would operate very badly in the broadband regime.
- the SEIRA signal is maximal when the radiative decay rates: $\gamma_r = (\omega_r - \omega_b)$. This demonstrates that the SEIRA signal is maximal for a resonator highly over-coupled ($f = \frac{\gamma_r}{\gamma_{nr}} \gg 1$).

Eventually, to simplify this discussion, we provide an intuitive picture of the mechanism behind the enhancement of the SEIRA signal. We show that the absorber induces a modification of the frequency $\omega_r = \omega_r + \omega_\mu$ and of the nonradiative decay rate $\gamma_{nr} = \gamma_{nr} + \gamma_\mu$ of the resonator that brings it back towards resonance and critical coupling.

Actions taken by the authors:

- We added a complete section in the Supplementary Information (Section 3) that contains 3 figures and the analytical study of the reflection coefficient in presence of the absorber using temporal coupled mode theory, and demonstrate that the SEIRA signal is maximal for a resonator highly over-coupled $f \gg 1$. We added a paragraph in the revised manuscript with equations 1 and 2 and added Fig. 2 to summarize the result of the derivation.

Comment 5: *Why is the electromagnetic near-field not maximal for the critical coupling case (as expected from theory and literature)?*

Response: *The definition of critical coupling ($R=0$) is not equivalent to maximum electromagnetic near field. Critical coupling entails a large enhancement for metals that are almost lossless. Here, as we show in Fig. S2, the maximum electromagnetic near field is obtained in the over-coupling regime ($f = 3$) when the slit geometry (h_s, w_s) is kept constant as well as the resonance wavelength.*

Action taken in the manuscript:

- We have added a discussion in Sec. 1.3 of the supplemental material.

Reviewer 2 :

General comment:

"Paggi et al. investigates broadband, high-contrast SEIRA sensing of thin films in a previously overlooked 'over-coupled' regime.

In the majority of SEIRA studies reported to date, as the authors correctly state, a common design approach has been to enhance IR absorption in bare resonator/antenna structures and then to insert target analytes in and around optical hotspots.

The authors approach is different in that they utilized an over-coupled resonator structure consisting of an array of gold ribbons coupled to a gold mirror. While the reflectivity spectrum of the bare resonator shows weak IR absorption, interestingly the structure shows broadband SEIRA signals from deposited PMMA and DNT thin films. This counter-intuitive outcome is nicely explained by using analytical equations and computer simulations.

The manuscript is clearly written and suggests a promising new approach for the SEIRA sensing community, so I recommend publication in Nature Communications after the authors address the following comments:"

Response: We are pleased that the referee finds our work timely and of interest for Nature Communication's readership. We are grateful for the positive evaluation on the scientific soundness and accessibility of our manuscript. We acknowledge the referee's constructive remarks which we address in detail below.

Comment 1: Before delving into the discussion on the critical vs. over-coupled regimes, it'd be nice to add more materials to explain the 'mechanics' of their resonator structure and how the geometrical parameters (gap width, ZnS thickness, ribbons periodicity, etc.) influence the overall IR absorption and transition from under- to critical- and over-coupled regimes.

The basic geometry of their resonator structure is reminiscent of a patch antenna structure. Also, similar motifs have been used by other researchers (albeit mostly in critically coupled regime), for example, the gapped gold antennas on a reflector [L. Dong et al. *Nano Lett.* 2017, 17, 5768-5774] and reflector-coupled gold ribbon arrays separated by a mid-IR transparent cavity [I.-H. Lee et al. *Nature Nanotechnology* 2019, 14, 313]. Thus, it would be nice to present the authors' design in the context of such relevant previous works.

Response: We thank the referee for pointing out the lack of this discussion on the resonator itself. We have added a discussion and a parametric study into the supplemental material (Sec. 1.2) to give insights of the behavior of the resonator. The resonance mechanism is pretty different from the two antennas of the publications given by the referee, even if they are relying on metal-insulator-metal stacks. In Dong et al. paper, the resonator acts as a bow-tie, and in Lee et al. paper, the resonator is a gap-plasmon cavity with a graphene layer.

In our case, our resonator acts as an optical Helmholtz-like resonance (See P. Chevalier et al., *Appl. Phys. Lett.* 2014, Chevalier, *Phys. Rev. B* 2014 and P. Chevalier et al., *Appl. Phys. Lett.* 2018). We provide a comparison of the behavior of the two configurations of optical Helmholtz resonators in the following figure (Fig. S2). We must emphasize that one important property of this resonator is to concentrate the electric field in the slit, while the magnetic field is concentrated in the dielectric layer.

Action taken by the authors:

- We have added in the supplemental material detailed information on the mechanism of resonance (Sec. 1.2 of the sup. Mat and Fig. S2).

Comment 2: *If it makes sense to improve the flow, the author might consider moving the field enhancement simulation (Figure S1) into the main text.*

Response: We have followed the suggestion of the reviewer in our revised manuscript

Action taken by the authors:

- The field enhancement figure is now included in the Fig. 1 of the manuscript.

Comment 3-i: *Figure 3 PMMA results:*

- Labels 'a', 'b', 'c', 'd' are mentioned in the caption but are missing in the figure. Please add proper labels.

Response: We thank the referee for pointing out this. We have corrected this in our revised manuscript.

Comment 3-ii:

- In (a) schematic, the authors depict ideal coverage of 45-nm thick PMMA on the film and inside the gap, which is unrealistic. Has the authors considered non-uniform coverage of the structure after PMMA coating? Could there be more (thicker) PMMA in and around the slit area?

Response: We do agree that the coverage of PMMA depicted is ideal. Usually, a spin-coated resist topography is influenced by the relief of lower layers.

We have done the following to try to assess the uniformity of the PMMA on the resonator:

- ellipsometric measurement of the thickness of PMMA near the resonators.
- AFM measurements on 3 periods of the resonator (1x1 μm^2 scanned area) at different locations and for 2 PMMA thicknesses.
- Infrared measurements on different spots (50 μm diameter) to assess the homogeneity of the infrared response.

We have added error bars in Fig. 4 of the main manuscript, we think that the homogeneity issue raised by the referee is partially responsible for the observed dispersion of the measurements. The infrared measurements are averaging the results on nearly 150 periods but there are still variations on the surface (the thickness of a spin-coated layer tends to be thicker near the edges of the sample). Ellipsometric measurements have shown that we typically had a 2 nm variation on the same sample.

Action taken by the authors:

- Sec. 2 of the supplemental material has been added with Figs. S3 and S4.
- Error bars on Fig. 4 (that now includes experimental results) account for the variations of homogeneity.

Comment 3-iii:

- More experimental spectra could be shown in Figure 3.

Response: We have added more experimental data points for various thicknesses of PMMA and DNT (see Figs. 4 and 5). See also our response to the next comment for the DNT measurements.

Action taken by the authors:

- Figs. 4 and 5 of the revised manuscript include more experimental data (as well as direct comparison with electromagnetic computations).

Comment 4: *Likewise, when the authors discuss the SEIRA results obtained from DNT in Fig. 4, I was hoping to find more details on how they prepared and characterized the sample after depositing a droplet. In the Supplementary Information, the authors infer that ~100 nm thick DNT film should have been formed, but more independent measurement would be desirable. Was the coverage very uniform across the active surface area?*

Response: We have given more details on the preparation and characterization of the sample. In fact, DNT is a volatile compound and very thin layers evaporate in a couple of minutes. This makes it hard to provide independent characterization of the thickness of DNT as could be done in the case of PMMA. In our revised manuscript, we have introduced additional measurements for two distinct times (one minute apart) and with a better control of the original deposited volume thanks to a microplotter device.

First, the microplotter allows us to deposit a controlled volume of a diluted DNT in acetonitrile on a controlled area of $500 \times 500 \mu\text{m}^2$ (with a typical lateral resolution of $10 \mu\text{m}$). The volume deposited is measured by the difference in the filling level of the capillary measured by a camera. The acetonitrile is evaporated in typically one minute, and we have been able to make infrared measurements at 3 and 4 minutes. One minute later, all the DNT is evaporated and there is no signal remaining.

Action taken by the authors:

- Sec. 2.2 of the supplemental material describes the process for deposition of DNT.
- New measurements have been included in Fig. 5.

Comment 5: *Could the authors discuss strategies to further optimize the performance? Their simulations show the E-field intensity enhancement ($|E|^2$) of 300x in their structure. By shrinking the gap width, it may be possible to further enhance the field enhancement factor. Could that lead to improved SEIRA sensitivity/contrast in the over-coupled regime?*

Response: We thank the reviewer for this suggestion. Indeed, shrinking the dimension of the slit width can lead to better performances of the resonator. We illustrate it with electromagnetic computations: if we consider a 10 nm slit (10 nm width and 8 nm height), performances obtained for smaller thicknesses are improved. As discussed in comment 1, the resonator acts as an optical Helmholtz resonator, and the slit acts as a capacitance. Reducing the width leads to higher E-field enhancements.

Action taken by the authors:

- Sec. 1.4 includes an example of electromagnetic simulations with a smaller slit resonator.

Comment 6: *In their 'Conclusion', it'd be nice to go beyond their current structure and discuss if the strategy of performing SEIRA in the over-coupled regime can be generalized. For example, could the authors suggest general design rules for other SEIRA structures (e.g. nanorods) based on what they've learned here?*

Response :

In response to both this comment of Reviewer 2 and comments of Reviewer 1, we have included both in the revised manuscript and the supplemental material an analytic model based on the temporal coupled mode theory. It gives a more general framework for using the over-coupled regime regardless of the resonator or the molecule considered. As shown in the new Fig. 2 of the manuscript, depending on the system (resonator + molecules) considered, there is an (over-)coupling ratio ($f > 1$) that is leading to a maximum SEIRA signal.

Actions taken by the authors:

- We added a complete section in the Supplementary Information (Section 3) that contains 3 figures and the analytical study of the reflection coefficient in presence of the absorber using temporal coupled mode theory, and demonstrate that the SEIRA signal is maximal for a resonator highly over-coupled $f \gg 1$. We added a paragraph in the revised manuscript with equations 1 and 2 and added Fig. 2 to summarize the result of the derivation.

Comment 7: In Figure 4, reflectivity different (y-axis), is the unit supposed to be %?

Response: We thank the reviewer for pointing out this. In fact, the reflectivity difference (as the reflectivity itself) can be expressed in percent (ranging from 0 to 100%) or can be directly expressed as a value from 0 to 1. Both are equivalent. Some authors are using the normalized reflectivity difference $\frac{\Delta R}{R}$ that must be expressed in %.

REVIEWER COMMENTS

Reviewer #1 (Remarks to the Author):

The manuscript has been improved in some respects (for example, by inclusion of additional experimental data), but some aspects still need more clarification:

(a) Page 3, "previous demonstrations (...) are still plagued by the SPR effect". SPR-based biospectroscopy is extremely established and methodologically sound. The simple fact that the authors' design may not require the same amount of baseline correction is not a crucial advance. The authors should tone down their language when making these comparisons. This should also be reworded on page 7 starting on line 145. In realistic sensing experiments (e.g., targeting biomolecules in more complex matrices), data from the authors' design will likewise require smoothing, baseline correction, etc.

(b) Page 3, the authors write "and report reflectivity differences up to 0.7 which also reproduce the molecular absorption spectrum". Stating reflectance differences without the corresponding film thickness makes no sense. The exact thickness of the film needs to be given to substantiate these claims.

(c) Page 11, "This detection responsivity could also be improved by reducing the width of the slits, but it would imply a trade-off between the interaction volume and the enhancement bandwidth as well as changes in the fabrication process. An example of design is given in Supplementary Fig. 3." The design given in the response letter and the SI is completely unrealistic and most likely impossible to realize in experiments. The percolation threshold of gold is in the 5-8 nm range, casting doubt on whether a continuous film of gold can actually form to make the ribbons. Furthermore, the gap size of 10 nm is close (or in fact below) the ultimate limits of electron beam lithography. Such extreme designs can and should not be used to quantify the performance of actual sensors. I suggest the authors either redo these simulations for a realistic set of geometrical parameters or remove them entirely. Consequently, the sensitivity claims should be checked and reduced if necessary throughout the manuscript.

(d) The DNT demonstration is convincing, but it is surprising that the "without DNT" curves show a noisy baseline with modulations, residual vibrational bands, etc., whereas for the PMMA results in Figure 3, the "no PMMA" curve is extremely clean and appears smoothed/processed in comparison. Also, a clear dip appears in the "no PMMA" data around 5.8 micron, which is absent (or at least significantly weaker) in the "without DNT" curve. The authors should carefully check how their data was processed, make sure that the experimental curves throughout the manuscript are processed in a similar fashion, and provide additional details on the data processing in the methods. Significant differences between data sets (as raised in the paragraph above) should be discussed and explained in the manuscript text.

Despite these further criticisms, I believe the manuscript has overall been improved and could be suitable for publication in Nature Communications once the above issues have been addressed.

Reviewer #2 (Remarks to the Author):

The authors have thoroughly revised the manuscript and provided detailed responses and actions taken in response to both referees' comments.

I recommend publication of this paper for Nature Communications.

Reviewer 1

Comment 1: *Page 3, "previous demonstrations (...) are still plagued by the SPR effect". SPR-based biospectroscopy is extremely established and methodologically sound. The simple fact that the authors' design may not require the same amount of baseline correction is not a crucial advance. The authors should tone down their language when making these comparisons. This should also be reworded on page 7 starting on line 145. In realistic sensing experiments (e.g., targeting biomolecules in more complex matrices), data from the authors' design will likewise require smoothing, baseline correction, etc.*

Response: We agree with the reviewer that SPR-based biospectroscopy is well established and it was not our intent to criticize it. Accordingly to the reviewer's comment, we have reworded the sentences in our manuscript so as to be more neutral.

Action taken in the manuscript:

The sentences have been reworded as follow.

Page 3: « Designs to enhance a larger portion of the spectrum with multi-resonant structures that could work with a single-pixel detector have also been proposed [23, 27, 33–37], but most of them have achieved weak SEIRA signals compared to previous demonstrations or signals resulting from coupled SEIRA and SPR effects. The integration of a broadband epsilon near zero material has also been suggested theoretically, but epsilon near zero effects have been so far experimentally demonstrated on very narrow spectral ranges [38]. Designing broadband sensitive SEIRA detectors is still an outstanding issue. »

Page 7: « Usually, SEIRA effect is quantified with post-treatment, using for instance smoothing algorithm or artificial materials to decouple SEIRA from SPR effect [13, 31]. Remarkably, in the over-coupled configuration, the SPR shift has less weight on reflectivity difference spectrum $\Delta R = R_{\text{without PMMA}} - R_{\text{with PMMA}}$ which therefore relates more directly to the molecule's signature and can be used to identify and quantify the chemical species. »

Comment 2: *Page 3, the authors write "and report reflectivity differences up to 0.7 which also reproduce the molecular absorption spectrum". Stating reflectance differences without the corresponding film thickness makes no sense. The exact thickness of the film needs to be given to substantiate these claims.*

Response: The sentence has been rewritten to specify the film thickness: « We demonstrate experimentally the ability to enhance the absorption of many lines over a broad spectrum from 5 to 10 μm which reproduces the molecular absorption spectrum and report reflectivity differences up to 0.7 for 45 nm thick PMMA film and 0.25 for 9 nm thick film. »

Comment 3: *Page 11, "This detection responsivity could also be improved by reducing the width of the slits, but it would imply a trade-off between the interaction volume and the enhancement bandwidth as well as changes in the fabrication process. An example of design is given in Supplementary Fig. 3." The design given in the response letter and the SI is completely unrealistic and most likely impossible to realize in experiments. The percolation threshold of gold is in the 5-8 nm range, casting doubt on whether a continuous film of gold can actually form to make the ribbons. Furthermore, the gap size of 10 nm is close (or in fact below) the ultimate limits of electron beam lithography. Such extreme designs can and should*

not be used to quantify the performance of actual sensors. I suggest the authors either redo these simulations for a realistic set of geometrical parameters or remove them entirely. Consequently, the sensitivity claims should be checked and reduced if necessary throughout the manuscript.

Response: Indeed, the proposed design represents a technological challenge as it requires to use e-beam lithography at its limit. One could use commercial high-resolution e-beam negative resists as HSQ or SX AR-N 8200 (Medusa 82) by ALLRESIST that has been demonstrated to achieve 10 nm resolution. But we acknowledge that this limit could be hard to reach and we have modified Figure S3 with 20 nm x 20 nm gap geometry.

Action taken in the manuscript:

To address the reviewer's concern, the Figure S3 in Supplementary has been modified by a

geometry taking into account 20 x 20 nm² slit sections which would be more accessible. The array is covered with an 8 nm thick layer of PMMA. The reflectivity difference demonstrates a 0.48 reflectivity difference for the 5.8 μm mode while maintaining broadband enhancement.

Comment 4: The DNT demonstration is convincing, but it is surprising that the "without DNT" curves show a noisy baseline with modulations, residual vibrational bands, etc., whereas for the PMMA results in Figure 3, the "no PMMA" curve is extremely clean and appears smoothed/processed in comparison. Also, a clear dip appears in the "no PMMA" data around 5.8 micron, which is absent (or at least significantly weaker) in the "without DNT" curve. The authors should carefully check how their data was processed, make sure that the experimental curves throughout the manuscript are processed in a similar fashion, and provide additional details on the data processing in the methods. Significant differences between data sets (as raised in the paragraph above) should be discussed and explained in the manuscript text.

The authors thank the reviewer for pointing at the missing information. Because of evaporation, the measurements on DNT molecules had to be acquired faster than the previous experiments. The number of averaged scans was reduced to 32 (instead of 64) while keeping a small diaphragm compared to the previous measurements on PMMA films which

are stable in time. The spectrum is therefore noisier. To be consistent, the measurement on the bare resonator was made with the same parameters.

The first bare array used for PMMA experiments in Fig 2.b. indeed presents a small peak at 5.8 μm before spin-coat of PMMA layer as analyte. This is due to the presence of residual traces of PMMA used for the lift off in the active volume of the slit. The mode is enhanced by the resonator.

It is not as visible in the last experiment with DNT as an additional in situ ion etching step was added in the fabrication process before deposition of the gold top layer to reduce traces of resist (and improve the quality of the lift off).

This information was added to the manuscript.

Reviewer 2

General comment: The authors have thoroughly revised the manuscript and provided detailed responses and actions taken in response to both referees' comments. I recommend publication of this paper for Nature Communications.

REVIEWERS' COMMENTS

Reviewer #1 (Remarks to the Author):

I am satisfied with the revisions and now support publication of the manuscript without further changes.

Reviewer 1

General comment: I am satisfied with the revisions and now support publication of the manuscript without further changes.

Response: We thank the reviewer for her/his helpful comments throughout the reviewing process. The input of both referees has been invaluable for improving the quality of the original manuscript and we thank them for their careful consideration.